# Concurrent Validity and Inter-Rater Reliability Assessment of Two Mental Toughness Instruments in Firefighting: A Two-Wave Longitudinal Study

**DOI:** 10.3390/bs14070523

**Published:** 2024-06-22

**Authors:** Andreas Stamatis, Zacharias Papadakis, Peter Beitia

**Affiliations:** 1Department of Health & Sport Sciences, University of Louisville, Louisville, KY 40292, USA; 2Sports Medicine Institute, University of Louisville Health, Louisville, KY 40208, USA; 3Department of Health Promotion and Clinical Practice, Barry University, Miami Shores, FL 33161, USA; zpapadakis@barry.edu (Z.P.); pbeitia369@gmail.com (P.B.)

**Keywords:** psychometrics, firefighter, mentally tough, tactical

## Abstract

*Mental toughness* (MT), a construct linked to high-stress performance, is predominantly studied via self-assessment in cross-sectional designs. In the firefighting context, where no dedicated MT instrument exists, the Sports Mental Toughness Questionnaire (SMTQ) and the Military Training Mental Toughness Inventory (MTMTI) have been used. However, their reliability and concurrent validity remain unexamined. This study employs a two-wave longitudinal design involving sixty-two male firefighters. Over two days, they completed the SMTQ, while colleagues and officers assessed their MT using the MTMTI. Analyses included concurrent validity and inter-rater reliability tests. Inter-rater reliability exhibited minimal agreement (Day 1: *κ* = 0.04, *p* = 0.172; Day 2: *κ* = 0.05, *p* = 0.063), low internal consistency (Day 1: *α* = 0.03, *ω* = 0.03; Day 2: *α* = 0.45, *ω* = 0.45), and weak inter-rater correlations (Day 1: ICC_2k_ < 0.001, 95%CI [−0.53, 0.35]; Day 2: ICC_2k_ = 0.13, 95%CI [−0.33, 0.43]). Concurrent validity showed limited correlations between self-assessed SMTQ scores and MTMTI ratings on both days (Day 1: *r* = −0.09, *p* = 0.5; Day 2: *r* = 0.1, *p* = 0.5). These findings may underscore the necessity to develop firefighting-specific MT tools, recognizing the unique demands and nuances of this high-stress profession.

## 1. Introduction

Firefighting is a high-stake, high-stress profession that demands not only physical preparedness but also psychological strength [1]. The nature of the job exposes firefighters to life-threatening situations, requiring them to operate at peak performance levels consistently [2]. This performance is not solely a function of physical capability; it is also deeply influenced by psychological factors that contribute to effective stress management and situational awareness [3].

One such psychological construct that has garnered attention in high-stress environments is *mental toughness* (MT), which encompasses the ability to thrive in challenging situations [4]. MT has been identified as a significant predictor of successful performance outcomes (e.g., sales, race time, GPA), equipping individuals with the mental skills (i.e., attention regulation, success mindset, facing adversity, buoyancy, context knowledge, emotion regulation, self-efficacy, optimistic style) to cope with adversity and make sound, goal-oriented decisions under pressure [5].

Despite the apparent importance of MT in the inherently stressful context of firefighting, there is a conspicuous absence of dedicated instruments designed to measure this construct within this specific setting [6]. Recently, Beitia et al. [6] employed the peer-rated Military Training Mental Toughness Inventory (MTMTI) [7] and the self-rated Sports Mental Toughness Questionnaire (SMTQ) [8] to assess MT within a firefighter cohort for two primary reasons: (a) the extant literature on MT is predominantly characterized by cross-sectional self-assessment methodological designs [9] and (b) the traumatic experiences encountered in military settings bear tactical similarities to those in firefighting [10].

Regardless of the MTMTI’s and SMTQ’s success in reliably capturing MT levels in their respective domains ([11]), these instruments were originally developed for military and sports settings, respectively. This raises concerns about their reliability and concurrent validity when applied to firefighting, a profession that also has unique demands and nuances from the military and athletic contexts (e.g., frequent immediate responsiveness, unique regulatory environment, community interaction/emotional toll; [12]).

The use of existing MT assessment tools, like the MTMTI and SMTQ, in firefighting is not merely a matter of convenience but a significant concern that warrants rigorous scientific investigation. The importance of psychometrically sound assessment tools cannot be overstated, especially when the stakes involve human lives and well-being [13]. Yet, there is a dearth of studies that examine the concurrent validity and inter-rater reliability of these MT assessment tools in the context of firefighting. This gap in the literature underscores the necessity for empirical scrutiny of these tools’ applicability and accuracy within this high-stress profession.

Against this backdrop, the aim of this study is to employ a two-wave longitudinal design to evaluate the concurrent validity of the SMTQ and MTMTI scores and the inter-rater reliability of MTMTI raters’ scores among a sample of firefighters. By rigorously testing these tools, this study seeks to address a significant gap in the existing literature and provide actionable insights for the development of more context-specific MT assessment tools. The findings have the potential to significantly enhance our understanding of MT assessment in firefighting and inform future tool development, thereby meaningfully contributing to performance optimization in this critical profession.

## 2. Materials and Methods

### 2.1. Study Design

This research was structured as a longitudinal study conducted in two waves, situated within an ongoing series of studies investigating the relationship between physiological and psychological attributes in relation to the Physical Ability Test (PAT) (for more details on the followed methodology, please see [6]). This longitudinal approach enabled a comparative analysis of MT at different time points while avoiding the common design in MT research (i.e., cross-sectional), thereby enriching the understanding of MT’s role in firefighters’ performance.

Ethical research practices were followed, with comprehensive participant briefings on the study’s scope, risks, and benefits, leading to the acquisition of informed consent in compliance with the IRB and the Declaration of Helsinki.

To enhance the methodological rigor of our investigation, we incorporated the National Heart, Lung, and Blood Institute’s (NHLBI) Quality Assessment Tool for Observational Cohort and Cross-Sectional Studies [14]. Established by the NHLBI in 2013, this suite of specialized quality assessment instruments is designed to aid scholars in appraising study designs for core elements crucial to internal validity. The suite includes a tool specifically crafted for the appraisal of observational cohort and cross-sectional studies, enabling a systematic evaluation of potential methodological flaws or implementation errors. The deployment of this instrument facilitated a meticulous quality assurance process in our research, ensuring adherence to stringent standards of internal validity. Table A1 (See Appendix A) delineates the individual criteria of the quality assessment tool and elucidates the application of each criterion within the context of our study.

### 2.2. Participants

The participant pool consisted of 62 male, active fire-suppression firefighters from two South Florida fire departments (FDs). Selection criteria included age (20 to 55 years) and active-duty status in fire-suppression roles. Health screenings were conducted to ensure that all participants were fit for duty. Recruitment took place from October 2020 to October 2022. In the development of the study protocol, we established the inclusion and exclusion criteria (e.g., not cleared to perform their occupational duties) a priori to ensure a systematic approach to participant selection. This pre-selection strategy was essential for maintaining methodological integrity and for the homogeneity of the study population. The criteria were meticulously designed to align with the research objectives and were uniformly applied to all potential subjects to guarantee that each participant met the same standards for inclusion. This uniform application of criteria was pivotal in minimizing selection bias and enhancing the validity of the study’s findings.

### 2.3. Sample Size Justification

In this research, the selection of the sample was based on the cohort sizes from two distinct FDs (convenience-based sample). An a priori analysis was performed using the *meddecide* module within *Jamovi*. This analysis aimed to determine the required number of subjects for an outcome level of 2, specifically focusing on Agreement vs. No Agreement, with 2 raters. The hypothesis testing for kappa (*κ*) levels of agreement involved H0 = 0.01, indicating no agreement, and HA = 0.9, signifying almost perfect agreement. Additionally, the prevalence of the trait was set at 0.50 [15]. The outcome of this analysis revealed that a minimum of 10 subjects is necessary for inter-observer agreement.

For the determination of concurrent validity and the sample size calculation based on Pearson’s r using GPower 3.1, an a priori sample size was established. This sample size calculation was conducted for two tails, where H0 represented no correlation, and HA represented a correlation of 0.99. The significance level (*α*) was set at 0.05, and the desired statistical power (1 − *β*) was set at 0.8. The outcome of this calculation indicated that a total sample size of 6 was required for this analysis.

### 2.4. Instruments

#### 2.4.1. Military Training Mental Toughness Inventory (MTMTI)

The MTMTI (available at: https://dspace.stir.ac.uk/bitstream/1893/21572/2/MT_Final_R2.pdf (accessed on 1 May 2024) (For further details on the psychometric soundness of the instruments, readers are encouraged to begin with the information provided in [11] and the provided links.) is a peer-rated tool comprising six items (e.g., “His recent performances have been poor”), specifically fashioned to gauge MT levels in military contexts. Adapted for our study, each firefighter’s MT was assessed by a superior officer and an additional evaluator from the same department, ensuring a dual perspective and enhancing the reliability of the measurement. Assessors rated responses, reflecting how firefighters cope with stressors like difficult conditions or criticism, on a seven-point Likert scale. The MTMTI’s comprehensive scoring encompasses factors such as confidence and resilience, providing an aggregate measure of a firefighter’s psychological readiness. In the existing body of literature on the military, the MTMTI has demonstrated robust psychometric properties, such as internal consistency scores, affirming its promising psychometric soundness for the intended evaluative purposes ([7,16]).

#### 2.4.2. Sports Mental Toughness Questionnaire (SMTQ)

The SMTQ (available at: https://www.researchgate.net/publication/344809575_The_relationship_of_physical_activity_and_mental_toughness_in_collegiate_esports_varsity_student-athletes/figures?lo=1 (accessed on 1 May 2024) (For further details on the psychometric soundness of the instruments, readers are encouraged to begin with the information provided in [11] and the provided links.), a 14-item self-assessment tool (e.g., “I am committed to completing the tasks I have to do”), gauges MT in athletes on a four-point Likert-type scale. Firefighters self-reported on attributes pertaining to confidence and constancy, such as their response to perceived threats and challenging situations. The total score derived from the SMTQ offers insight into three core components of MT: confidence, constancy, and control. In scholarly research related to sports, the SMTQ has demonstrated satisfactory psychometric properties, such as internal consistency scores, confirming its psychometric efficacy for the designated assessment objectives ([17,18]).

### 2.5. Procedure

The design facilitated examination of the concurrent validity of the SMTQ and MTMTI tools and the inter-rater reliability of MTMTI assessments within a firefighting context. Participants, after signing the informed consent, underwent the MT assessments twice (Influenced by the concept of “… synergy between substance and method.” [5] (p. 21), the assessment schedule for this study was meticulously designed. A minimum interval of 48 hours and a maximum of one week was maintained between assessments to minimize potential carry-over effects and to capture intra-individual variability. Furthermore, the scheduling was synchronized with the firefighters’ operational shifts (48 hours on, 96 hours off) to ensure feasibility and maximize participation in the study. Data collection was conducted within these specified shifts.) subsequent to completing the Physical Ability Test (PAT). Assessors were instructed to provide their evaluations immediately post-assessment to the research team for data integrity and confidentiality. Firefighters completed the SMTQ under the same conditions, ensuring data consistency for intra-rater reliability. The participation rate of eligible people was 100% and there was no loss to follow-up after baseline.

### 2.6. Statistical Analyses

Both the concurrent validity and the inter-rater variability statistical tests were conducted, using the *R* statistical packages within the *Jamovi* software version 2.4.8, adhering to a significance threshold of *p* < 0.05. Evaluation criteria included Cohen’s *κ,* where ≤ 0 indicates no agreement, 0.01–0.20 suggests none to slight agreement, 0.21–0.40 denotes fair agreement, 0.41–0.60 signifies moderate agreement, 0.61–.80 indicates substantial agreement, and 0.81–1 represents almost perfect agreement [19]. Additionally, Cronbach’s *α*, McDonald’s *ω,* and the interclass correlation coefficient (ICC) were employed, with cut-off points as follows: ≤0.5—unacceptable, ≥0.5—poor, ≥0.6—questionable, ≥0.7—acceptable, ≥0.8—good, ≥0.9—excellent [20,21,22,23,24].

#### 2.6.1. Inter-Rater Reliability

Cohen’s kappa (*κ*), Cronbach’s alpha (*α*), McDonald’s omega (*ω*), and the Intraclass correlation coefficient (ICC) were utilized to evaluate the consistency among raters of the MTMTI.

#### 2.6.2. Concurrent Validity

The Z-scores of the MTMTI and SMTQ were calculated for each assessment day to facilitate statistical comparison. Pearson correlation coefficients provided a measure of the concordance between self-assessed SMTQ scores and peer-rated MTMTI scores.

## 3. Results

### 3.1. Internal Validity

Based on the Quality Assessment Tool for Observational Cohort and Cross-Sectional Studies provided in Table A1 (see Appendix A), all applicable criteria (1–5, 11, 13) were satisfied. The non-applicability of certain criteria (6–10, 12, 14) due to the specific nature and focus of this study does not detract from its overall internal validity.

### 3.2. Inter-Rater Reliability of MTMTI

The inter-rater reliability results are summarized in Table 1 and illustrated in Figure 1. On Day 1, Light’s kappa (*κ*) was 0.05 (z = 1.86, *p* = 0.063), and on Day 2, it was 0.05 (z = 1.86, *p* = 0.063), indicating minimal agreement between raters. Figure 1 further supports this minimal agreement, as the plotted values demonstrate substantial variability and overlap, highlighting inconsistencies among the raters.

Additionally, Table 2 presents internal consistency measures. On Day 1, Cronbach’s alpha (*α*) and McDonald’s omega (*ω*) were both 0.03. On Day 2, *α* and *ω* improved to 0.45, but these values are still considered unacceptable, indicating that the MTMTI items do not consistently measure the same construct of mental toughness.

Figure 2 indicates that inter-rater correlations were weak, even when significant. Specifically, correlations of 0.29 and 0.31 were significant at *p* < 0.05, and a correlation of 0.43 was significant at *p* < 0.001. These figures support the conclusion that, while there were some significant correlations, the overall agreement among raters remained low.

### 3.3. Concurrent Validity of MTMTI and SMTQ

Figure 3 and Figure 4 present the concurrent validity results, underscoring the weak correlations and low reliability between the MTMTI and SMTQ. On Day 1, inventory scores were negatively correlated (*r* = −0.09, *p* = 0.5), while on Day 2, they were positively correlated (*r* = 0.1, *p* = 0.5). The reliability agreement on Day 1 was extremely poor (ICC_2k_ < 0.001, 95%CI [−0.53, 0.35]) (Figure 3). On Day 2, reliability improved but remained relatively low (ICC_2k_ = 0.13, 95%CI [−0.33, 0.43]) (Figure 4). Cohen’s *κ* for both days indicated no agreement between the inventory scores (*k* < 0.001).

## 4. Discussion

In this study, we sought to critically evaluate the inter-rater reliability of MTMTI raters’ scores and the concurrent validity of the SMTQ and MTMTI scores among a sample of firefighters using a two-wave longitudinal approach. Our study’s adherence to the rigorous criteria set forth by the Quality Assessment Tool for Observational Cohort and Cross-Sectional Studies underscores its high internal validity. We meticulously ensured that every applicable standard was met or exceeded, particularly in the definition and specification of the study population, clarity of the research objectives, and the consistent implementation of valid and reliable outcome measures. Such methodological precision, especially in the homogenous selection of participants and the consistent application of measurement tools, reinforces the legitimacy of our findings and their contribution to the understanding of *mental toughness* (MT) in firefighters.

In general, our findings reveal a notable disconnect between the use of those two inventories in the context of firefighting: (a) the inter-rater reliability of the MTMTI demonstrated minimal agreement and (b) the correlations between the SMTQ self-assessments and MTMTI peer assessments were weak and statistically insignificant on both assessment days.

### 4.1. MTMTI Rater 1 vs. MTMTI Rater 2

The psychometric evaluation of the MTMTI reveals significant issues. The minimal *κ* values along with the heatmap analyses suggest that raters do not consistently agree on the scores they assign, which is problematic for an inventory that should exhibit robust inter-rater reliability, especially when used in applied settings [15,19,25].

The internal consistency values are also troubling. An *α* or *ω* value below 0.7 is generally considered to be inadequate, and the reported values on Day 1 are exceptionally low [20,23,24]. While there is an improvement on Day 2, consistency remains below the desirable threshold. This suggests that the items within the MTMTI may not be cohesively measuring the same underlying construct of MT.

### 4.2. SMTQ vs. MTMTI

Upon integrating both qualitative observations from the scatterplots and the quantitative metrics, our analysis raises significant concerns. On Day 1, the near-zero negative correlation (*r* = −0.09) coupled with a non-significant *p*-value of 0.5 suggests an absence of a systematic relationship between the MTMTI and SMTQ scores [25]. This is supported by a scatterplot showing a wide dispersion of Z-scores for MTMTI. Moreover, the interclass correlation coefficient (ICC_2k_) being less than 0.001, with a wide confidence interval crossing zero, indicates an extremely poor reliability agreement among raters [22]. This is particularly troubling for a metric that requires high reliability to be deemed valid. Additionally, Cohen’s *κ* value near zero corroborates this lack of agreement, casting further doubt on the reliability of the scores of these measures.

Moving to Day 2, while there is a shift to a weak positive correlation (*r* = 0.1), it remains statistically insignificant. The slight improvement in reliability (ICC_2k_ = 0.13) is still substantially below acceptable thresholds, as evidenced by the confidence interval that spans into negative values [26]. This marginal enhancement in consistency is visually apparent in the lesser spread of Z-scores for MTMTI on the scatterplot. Nonetheless, the Cohen’s *κ* value continues to indicate no agreement between the inventory scores [15,19,25].

### 4.3. Future Studies

From an overarching psychometric perspective, our evaluation of these findings centers on the possibility for systematic bias in the measurement process [27]. Across both days, neither the reliability nor the inter-rater agreement reach the level expected for psychological assessments, casting doubts on the validity of score-based inferences. The persistent divergence between self-assessed SMTQ and peer-reviewed MTMTI scores justifies raising questions about their mutual measurement efficacy for MT facets. These discrepancies and low reliability levels suggest either a misalignment in construct capture or an inherent subjectivity and lack of control in the assessment process [28].

The enduring low agreement, despite some improvements, hints at systemic bias, potentially rooted in issues with the rating scales or rater training [27,29]. It becomes evident that these scales may not be optimally functioning in the current context, as their design was never tailored for it. Consequently, the development of a firefighting-specific instrument is our paramount recommendation for future studies, as it could address these repetitive reliability and agreement issues. As a secondary, albeit lesser, recommendation, we propose that enhancing rater education and training could be pivotal. This intervention could become critical, regardless of whether the existing scales are updated or a new, firefighter-specific scale is introduced. Notably, there appears to be a lack of shared understanding among raters regarding the construct being assessed with the SMTQ and MTMTI. Comprehensive rater training would be beneficial in both scenarios.

It is also pertinent to note that methodological triangulation/multi-informant assessment has been advocated as an effective approach for MT questionnaire utilization [9]. In this context, the creation of a new firefighter-specific questionnaire based on triangulation principles is advisable. In our study, we did meticulously control for extraneous variables using the NHLBI Quality Assessment Tool. However, one delimitation of our study was the variance in rater perspectives. This dichotomy (self- vs. peer assessment) in chosen assessment modalities may have inherently introduced variability in the interpretation of MT within the firefighting context. Rather than debating the merits/drawbacks of self- ([30]) *or* peer assessment ([31]), a combined self- *and* peer assessment approach is recommended for future studies ([32,33], as a potentially more holistic measurement framework [34].

### 4.4. Limitations

Although the vast majority of US firefighters are males (95.6%; [35]), our study’s limitations include a homogeneous all-male sample, which may limit the generalizability of our findings to broader north American firefighter populations. In addition, as a convenience sample, its confinement to a specific geographic region may not represent those in different regions or states. Furthermore, although the creators of the instruments do not prescribe specific training for the raters, one of the researchers provided an explanation of the inventories to the Chief, who then informed the firefighters. Consequently, more rigorous and detailed training could potentially have yielded more accurate MT scores. Lastly, due to the unavailability of comprehensive demographic information from the participating departments, we cannot ascertain that our sample accurately represents the entire firefighter populations of these departments.

## 5. Conclusions

The current study’s psychometric analysis casts substantial doubt on the suitability of the SMTQ and MTMTI instruments for evaluating MT in the firefighting context. The observed weak correlations and negligible inter-rater reliability underscore the need for a specialized, context-specific assessment tool that can accurately reflect the unique psychological landscape and professional demands faced by firefighters. Our findings indicate that the current instruments, developed within sports and military settings, do not succeed in capturing the nuanced facets of MT pertinent to firefighting. Consequently, the development of a bespoke measurement instrument, complemented by enhanced rater training, emerges as a critical step towards achieving reliable MT scores and valid inferences in this high-stakes profession. This study not only contributes to the existing body of psychometric MT research, but also underscores the essential role of tailored assessment tools in fostering a robust understanding of MT within specialized occupational domains.

## Figures and Tables

**Figure 1 behavsci-14-00523-f001:**
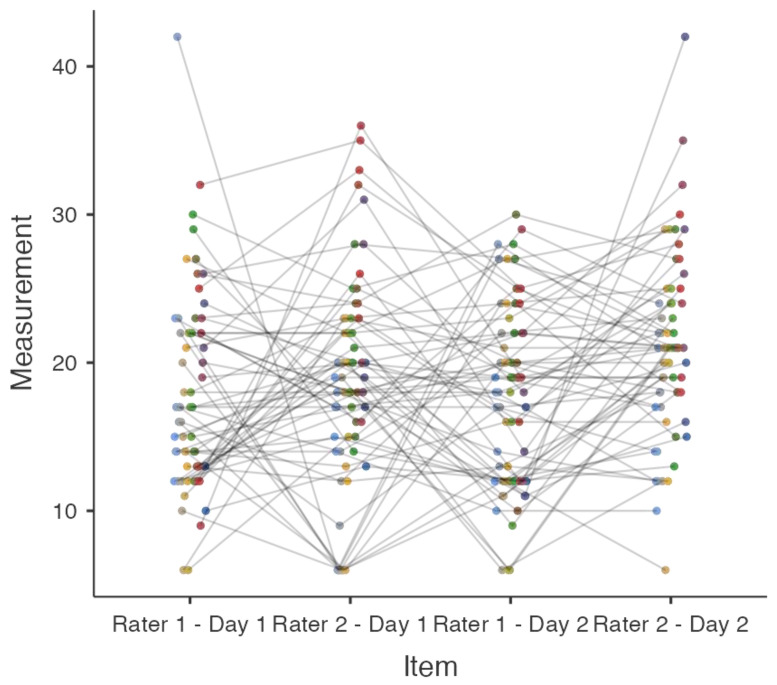
MTMTI reliability plot for Days 1 and 2 between raters. Different colored circles represent individual participants. Each color is assigned to a unique participant, allowing for a clear visual distinction between the data points corresponding to each participant’s responses on Days 1 and 2.

**Figure 2 behavsci-14-00523-f002:**
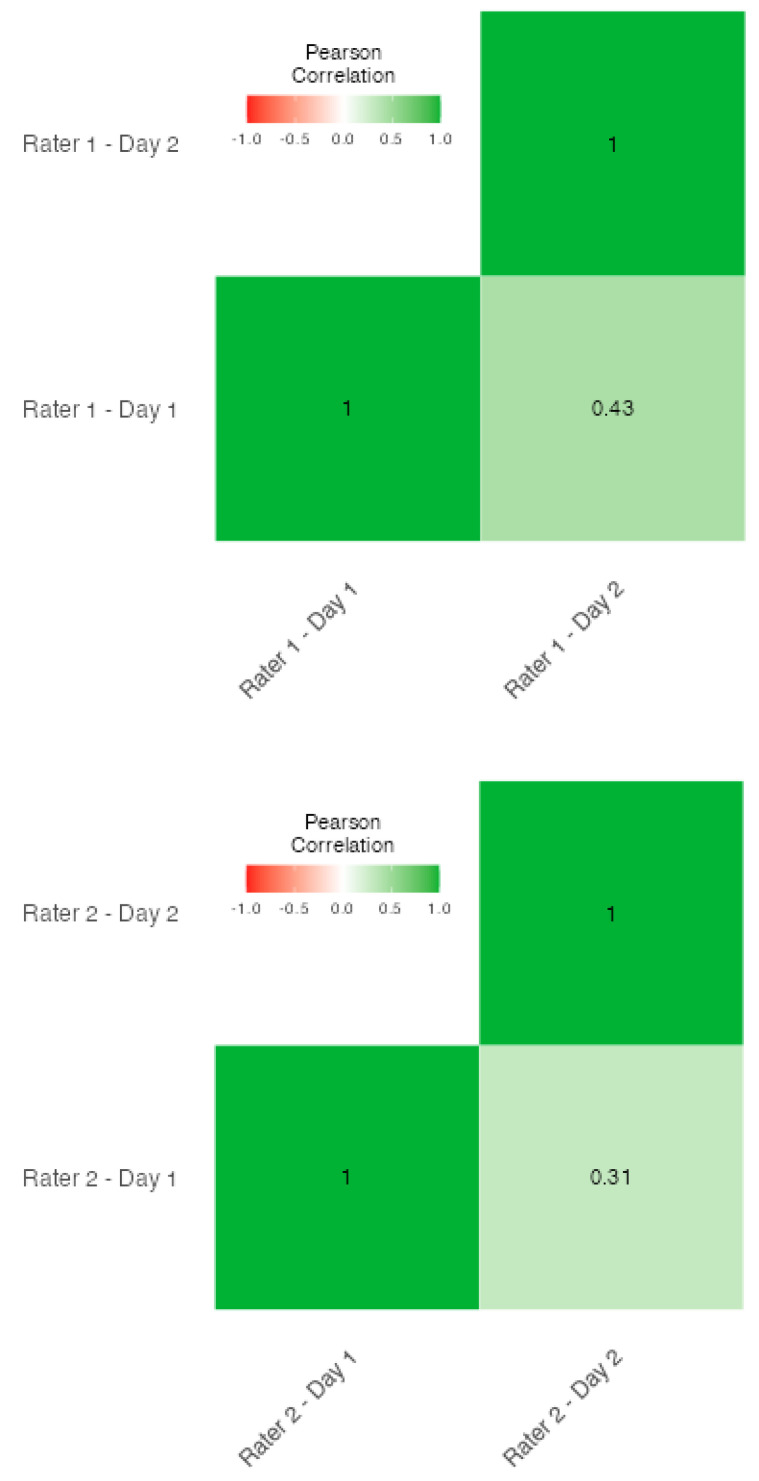
Correlation heatmaps between raters and days.

**Figure 3 behavsci-14-00523-f003:**
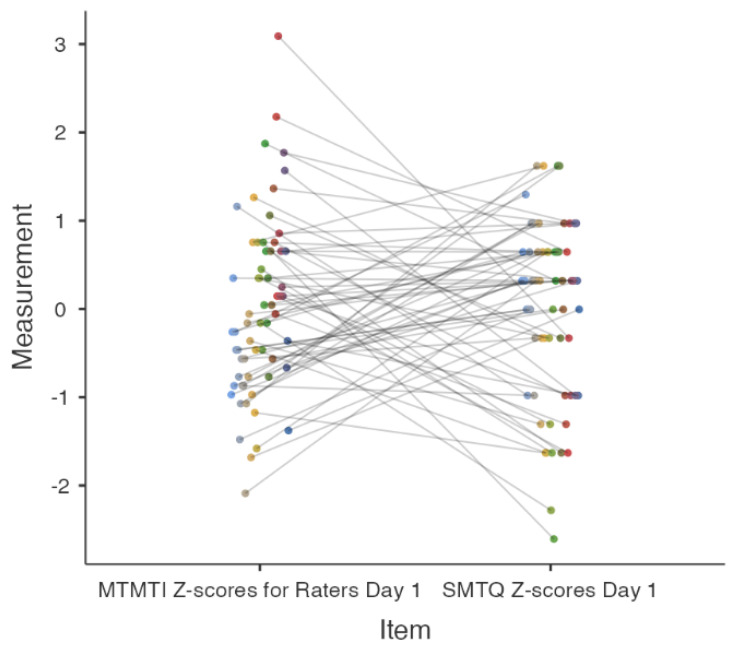
Reliability plot for MTMTI and SMTQ in Z-scores for Day 1. MTMTI = Military Training Mental Toughness Inventory. SMTQ = Sports Mental Toughness Questionnaire. Different colored circles represent individual participants. Each color is assigned to a unique participant, allowing for a clear visual distinction between the data points corresponding to each participant’s responses on Day 1.

**Figure 4 behavsci-14-00523-f004:**
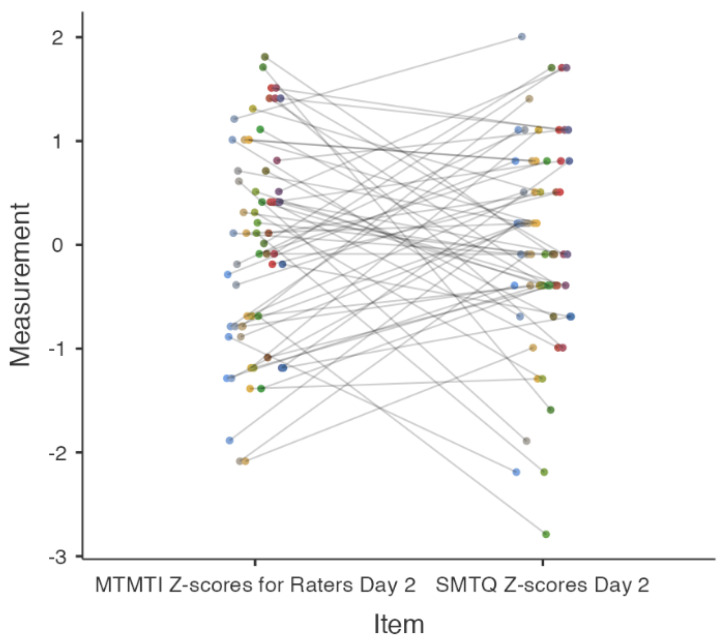
Reliability plot for MTMTI and SMTQ in Z-scores for Day 2. MTMTI = Military Training Mental Toughness Inventory. SMTQ = Sports Mental Toughness Questionnaire. Different colored circles represent individual participants. Each color is assigned to a unique participant, allowing for a clear visual distinction between the data points corresponding to each participant’s responses on Day 2.

**Table 1 behavsci-14-00523-t001:** Inter-rater reliability between raters and days.

	n	Rater	Statistic	z	*p*
Inter-rater reliability for Day 1					
Light’s kappa ^a^	62	2	0.05	1.86	0.063
Inter-rater reliability for Day 2					
Light’s kappa ^a^	62	2	0.05	1.86	0.063
Inter-rater reliability for Rater #1 for Day 1 vs. Day 2					
Light’s kappa ^a^	62	2	0.07	2.31	0.021
Inter-rater reliability for Rater #2 for Day 1 vs. Day 2					
Light’s kappa ^a^	62	2	0.07	2.67	0.008

^a^ Light’s kappa represents the average Cohen’s kappa when there are two or more raters.

**Table 2 behavsci-14-00523-t002:** Internal consistency measures between raters and days.

	Mean	SD	Cronbach’s *α*	McDonald’s *ω*
MTMTI Reliability Statistics—Raters by Day 1				
MTMTI	18.28	4.92	0.03	0.03
MTMTI Reliability Statistics—Raters by Day 2				
MTMTI	19.44	5.01	0.03	0.45
MTMTI Reliability Statistics—Days by Rater #1				
MTMTI	17.85	5.45	0.60	0.60
Inter-rater reliability for Rater #2 for Day 1 vs. Day 2				
MTMTI	19.86	5.43	0.47	0.48

Note. MTMTI = Military Training Mental Toughness Inventory.

## Data Availability

Data are available upon reasonable request from the corresponding author.

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
