# Peer review of "Concurrent Validity and Inter-Rater Reliability Assessment of Two Mental Toughness Instruments in Firefighting: A Two-Wave Longitudinal Study"

_behavsci, 2024, doi:10.3390/bs14070523_

Round 1

Reviewer 1 Report

Comments and Suggestions for Authors

Abstract

The term “psychometric soundness” is a very broad term. In this paper specifically, it is assessing reliability and concurrent validity of the measures. For clarity, I would recommend replacing “psychometric soundness” with those terms instead. 

1. Introduction

Overall, very well written; I only have a handful of suggestions to improve clarity.

Lines: 48-53

The entire paragraph should be reviewed and rewritten for clarity and intent. Right now, the purpose of the paragraph is lost.

Line 48-49: The sentence reads: “Regardless MTMTI’s and SMTQ’s success in their respective contexts (e.g., [11]), these instruments were originally developed for military and sports settings, respectively.” Please rewrite for clarity. Additionally, it is unclear what “success” means in this sentence; rephrasing is recommended. Also, “e.g.,” should be removed - if there is another citation that should include, please include it. As it is written, I am not sure why [11] is cited in this sentence.

2. Materials and Methods

2.2 Participants

Given the diversity within the firefighting profession, I recommend adding more information about the participants. For example, what is their shift schedule (e.g., 48-hours on, 96 off; Kelly schedule, etc.); approximately how many calls does the department run a year; how large is the department; how many years of experience do they have? Since these are volunteer firefighters, is there a minimum number of days they are required to volunteer? Or can one person volunteer for just one shift a year? More explanation is necessary especially because requirements of a volunteer firefighter and a full-time firefighter can be vastly different. So, it is possible that your sample was extremely diverse, and you were measuring MT of someone who volunteers a few days a year to a full-time firefighter. 

This information would be beneficial for future researchers – these variables are important to consider when conducting research on this population. I would also recommend adding a few sentences to the limitations about the population (especially if it is diverse).

2.4 Instruments

Are the MTMTI and SMTQ open-access instruments? If so, I would recommend including those as supplemental materials.

Line 133 & 142 – the “e.g.,” should be removed before the citations

2.6 Statistical Analyses

Line 161: The last sentence is not complete. It reads: “More specifically:” – it likely needs to be deleted.

2.6.1. Interrater Reliability

Four different statistics were conducted to assess interrater reliability. I would recommend more context as to why all of them are necessary. Also, it is not clear why these analyses were only conducted on the MTMTI and not the SMTQ.

Additionally, there is nothing about how raters were trained. This addition must be made to the paper especially because it is discussed at length in the discussion (as a limitation and future research suggestion). 

2.6.2 Concurrent Validity

Lines 170 – 181

I am assuming this needs to be deleted? It looks like instruction text.

4. Discussion

Line 292 & 293 - the “e.g.,” should be removed before the citations

4.4 Limitations

Have model or scale structure been assessed in either of the instruments? While I recognize that this may be outside the scope of the paper, this is a crucial piece that should occur before an instrument can be recommended for use in research or in practice. For example, have the instruments been assessed using confirmatory factor analysis, structural equation modeling, or invariance testing? Any information related to ensuring the scale is measuring what it is intending to measure should be added to the paper and/or added as a limitation and future research.

Author Response

We would like to thank the Reviewer for his time and energy. The revised version is of better quality thanks to their comments. 

Below, we provide a point-by-point response:

Abstract

The term “psychometric soundness” is a very broad term. In this paper specifically, it is assessing reliability and concurrent validity of the measures. For clarity, I would recommend replacing “psychometric soundness” with those terms instead. 

Thank you for this valuable suggestion. We replaced the term “psychometric soundness” with “reliability and concurrent validity” in the abstract to enhance clarity.

  1. Introduction

Overall, very well written; I only have a handful of suggestions to improve clarity.

Lines: 48-53

The entire paragraph should be reviewed and rewritten for clarity and intent. Right now, the purpose of the paragraph is lost.

Line 48-49: The sentence reads: “Regardless MTMTI’s and SMTQ’s success in their respective contexts (e.g., [11]), these instruments were originally developed for military and sports settings, respectively.” Please rewrite for clarity. Additionally, it is unclear what “success” means in this sentence; rephrasing is recommended. Also, “e.g.,” should be removed - if there is another citation that should include, please include it. As it is written, I am not sure why [11] is cited in this sentence.

We appreciate your feedback. We have revised the paragraph for clarity and intent. We have clarified the term "success" in the sentence and removed the "e.g.," to improve readability and accuracy of citation.

  1. Materials and Methods

2.2 Participants

Given the diversity within the firefighting profession, I recommend adding more information about the participants. For example, what is their shift schedule (e.g., 48-hours on, 96 off; Kelly schedule, etc.); approximately how many calls does the department run a year; how large is the department; how many years of experience do they have? Since these are volunteer firefighters, is there a minimum number of days they are required to volunteer? Or can one person volunteer for just one shift a year? More explanation is necessary especially because requirements of a volunteer firefighter and a full-time firefighter can be vastly different. So, it is possible that your sample was extremely diverse, and you were measuring MT of someone who volunteers a few days a year to a full-time firefighter. 

This information would be beneficial for future researchers – these variables are important to consider when conducting research on this population. I would also recommend adding a few sentences to the limitations about the population (especially if it is diverse).

Thank you for this comment!
The demographics of the departments were not made available to us; so, we have acknowledged this as a limitation in the manuscript. The shift schedule for participants was 48 hours on, 96 hours off, and this information has been added in Footnote 2. The participants were indeed active firefighters. To clarify, we used the term 'volunteers' to refer to their voluntary participation in the research project. To avoid confusion, we have replaced 'volunteers' with 'active fire suppression personnel' in the Participants section.

2.4 Instruments

Are the MTMTI and SMTQ open-access instruments? If so, I would recommend including those as supplemental materials.

Thank you! We added to links in the Instruments section to facilitate readers who would like to access the instruments. 

Line 133 & 142 – the “e.g.,” should be removed before the citations

Removed.

2.6 Statistical Analyses

Line 161: The last sentence is not complete. It reads: “More specifically:” – it likely needs to be deleted.

Deleted.

2.6.1. Interrater Reliability

Four different statistics were conducted to assess interrater reliability. I would recommend more context as to why all of them are necessary. Also, it is not clear why these analyses were only conducted on the MTMTI and not the SMTQ.

Thank you for the opportunity to clarify!

To thoroughly assess interrater reliability, we utilized four different statistical methods: Cohen’s kappa (κ), Cronbach’s alpha (α), McDonald’s omega (ω), and the Intraclass Correlation Coefficient (ICC). Each of these statistics offers unique insights into the reliability and consistency of the ratings provided by different raters, thus providing a comprehensive evaluation.

  1. Cohen’s Kappa (κ): This statistic measures the agreement between two raters, adjusting for agreement that could occur by chance. It is particularly useful for categorical data and provides an index that ranges from -1 to 1, where higher values indicate better agreement. In our study, Cohen’s kappa helps us understand the level of agreement between raters beyond what would be expected by chance alone.

  2. Cronbach’s Alpha (α): This statistic assesses the internal consistency of the ratings. It measures how closely related a set of items are as a group. A higher Cronbach’s alpha indicates that the items measure the same underlying construct. This is crucial for ensuring that the items within the MTMTI are cohesively measuring mental toughness.

  3. McDonald’s Omega (ω): Similar to Cronbach’s alpha, McDonald’s omega provides an estimate of the reliability of the scale. It is often considered a more accurate measure of internal consistency, especially in cases where the assumptions of Cronbach’s alpha are violated. Including McDonald’s omega allows for a more robust assessment of internal consistency.

  4. Intraclass Correlation Coefficient (ICC): The ICC assesses the reliability of measurements or ratings. It evaluates the consistency or conformity of measurements made by multiple raters measuring the same quantity. The ICC is particularly useful for continuous data and provides a nuanced understanding of interrater reliability, reflecting both consistency and agreement among raters.

These statistical methods collectively ensure a comprehensive evaluation of interrater reliability, capturing different dimensions of consistency and agreement. This multi-faceted approach strengthens the validity of our findings and ensures that the reliability of the MTMTI is thoroughly assessed.

Regarding the SMTQ, these analyses were not conducted because the SMTQ is a self-report instrument with only one rater—the participant themselves. Therefore, interrater reliability is not applicable to the SMTQ, as it does not involve multiple raters providing ratings on the same items. Instead, the focus for the SMTQ would be on measures such as internal consistency and test-retest reliability to assess its psychometric properties

Additionally, there is nothing about how raters were trained. This addition must be made to the paper especially because it is discussed at length in the discussion (as a limitation and future research suggestion). 

You are right. We added information in the Limitations section. 

2.6.2 Concurrent Validity

Lines 170 – 181

I am assuming this needs to be deleted? It looks like instruction text.

Deleted.

  1. Discussion

Line 292 & 293 - the “e.g.,” should be removed before the citations

Removed.

4.4 Limitations

Have model or scale structure been assessed in either of the instruments? While I recognize that this may be outside the scope of the paper, this is a crucial piece that should occur before an instrument can be recommended for use in research or in practice. For example, have the instruments been assessed using confirmatory factor analysis, structural equation modeling, or invariance testing? Any information related to ensuring the scale is measuring what it is intending to measure should be added to the paper and/or added as a limitation and future research.

We appreciate your insightful comment regarding the assessment of the model or scale structure for the instruments used in this study. As you noted, such analyses are indeed beyond the scope of our current manuscript. For readers seeking more detailed information on the reliability and validity of the scores, we refer them to Citation 11 and the two added links, which offers a solid starting point for further exploration (see Footnote 1).

Given that we do not encourage the future use of these instruments in firefighting contexts, we emphasize the need for the development of a more appropriate inventory tailored specifically to the unique demands of this profession. This recommendation has been included in the Limitations section and highlighted as a potential direction for future research.

Reviewer 2 Report

Comments and Suggestions for Authors

Efforts to validate tools to measure skills or virtues important to firefighters are commendable. However, improvement is needed in writing the paper. What I think would like to be improved is as follows.

1. Please present the concept of mental toughness and its effects in more detail in the introduction.

2. It would be good to emphasize that mental toughness is different from psychological hardiness.

3. There is no need to present descriptive statistics in the abstract.

4. Please do not simply list the tables and figures in 3.2 and 3.3, but provide detailed explanations about them.

5. Psychometric findings are not properly discussed.

6. In addition to the limitations of the study currently described, there are additional limitations to this study. Please add it.

Author Response

We would like to thank you the Reviewer for their time and energy. This revised version is better thanks to their recommendations! 

Below, we provide point-by-point response:

Efforts to validate tools to measure skills or virtues important to firefighters are commendable. However, improvement is needed in writing the paper. What I think would like to be improved is as follows.

1. Please present the concept of mental toughness and its effects in more detail in the introduction.

Thank you for this comment. We added more information in Paragraph #2.

2. It would be good to emphasize that mental toughness is different from psychological hardiness.

Thank you for your suggestion. Mental toughness is indeed different from many constructs. Since psychological hardiness is not relevant to our research focus, we have chosen not to highlight this particular comparison in our manuscript.

3. There is no need to present descriptive statistics in the abstract.

Thank you for your feedback. We respectfully disagree, as we believe reporting the sample size in the abstract is important. It provides context and implies the power of our statistical findings. Therefore, we will retain this information in the abstract.

4. Please do not simply list the tables and figures in 3.2 and 3.3, but provide detailed explanations about them.

Thank you for this comment. We added more information, especially in 3.2.

5. Psychometric findings are not properly discussed.

Thank you for your comment. We are unclear on the specific concerns being raised. Based on our interpretation and feedback from the other Reviewer, we believe the psychometric findings are adequately discussed. If you could provide more detailed information or specific points of concern, we would be happy to address them in greater detail.

6. In addition to the limitations of the study currently described, there are additional limitations to this study. Please add it.

 Thank you for this comment. We added more limitations in that section.

Round 2

Reviewer 1 Report

Comments and Suggestions for Authors

All of my comments were adequately addressed and I believe this version of the paper can be accepted in its present form.

Author Response

Thank you for your time, energy, and insightful comments!

Reviewer 2 Report

Comments and Suggestions for Authors

I think you have made many improvements to the points pointed out in the first review.

However, if you think a little more about what I commented and supplement it by carefully examining other details, I think the paper will be of higher quality.

Author Response

Dear Reviewer,
Thank you for your positive feedback and for acknowledging the improvements made to our manuscript.

We agree that further enhancements are always possible. However, we are currently uncertain about the specific details you are referring to in your latest comments, making it challenging for us to address them effectively.

We appreciate your insights and the time you've invested in reviewing our paper.

Best regards,
The Authors.